# Clinically Applicable Quantitative Magnetic Resonance Morphologic Measurements of Grey Matter Changes in the Human Brain

**DOI:** 10.3390/brainsci11010055

**Published:** 2021-01-05

**Authors:** Tong Fu, Xenia Kobeleva, Paul Bronzlik, Patrick Nösel, Mete Dadak, Heinrich Lanfermann, Susanne Petri, Xiao-Qi Ding

**Affiliations:** 1Institute of Diagnostic and Interventional Neuroradiology, Hannover Medical School, 30625 Hannover, Germany; Bronzlik.Paul@mh-hannover.de (P.B.); Noesel.Patrick@mh-hannover.de (P.N.); m.dadak@vincenz.de (M.D.); Lanfermann.Heinrich@mh-hannover.de (H.L.); ding.xiaoqi@mh-hannover.de (X.-Q.D.); 2Department of Neurology and Neurophysiology, Hannover Medical School, 30625 Hannover, Germany; xkobelaeva@posteo.de (X.K.); Petri.Susanne@mh-hannover.de (S.P.); 3Department of Neurology, University of Bonn, 53127 Bonn, Germany

**Keywords:** CAT12 toolbox, clinically applicable qMRI brain grey matter measurement, healthy ageing brain grey matter, neurodegenerative disease diagnosis, ALS

## Abstract

(1) Purpose: Quantitative magnetic resonance imaging (qMRI) measurements can be used to sensitively estimate brain morphological alterations and may support clinical diagnosis of neurodegenerative diseases (ND). We aimed to establish a normative reference database for a clinical applicable quantitative MR morphologic measurement on neurodegenerative changes in patients; (2) Methods: Healthy subjects (HCs, *n* = 120) with an evenly distribution between 21 to 70 years and amyotrophic lateral sclerosis (ALS) patients (*n* = 11, mean age = 52.45 ± 6.80 years), as an example of ND patients, underwent magnetic resonance imaging (MRI) examinations under routine diagnostic conditions. Regional cortical thickness (rCTh) in 68 regions of interest (ROIs) and subcortical grey matter volume (SGMV) in 14 ROIs were determined from all subjects by using Computational Anatomy Toolbox. Those derived from HCs were analyzed to determine age-related differences and subsequently used as reference to estimate ALS-related alterations; (3) Results: In HCs, the rCTh (in 49/68 regions) and the SGMV (in 9/14 regions) in elderly subjects were less than those in younger subjects and exhibited negative linear correlations to age (*p* < 0.0007 for rCTh and *p* < 0.004 for SGMV). In comparison to age- and sex-matched HCs, the ALS patients revealed significant decreases of rCTh in eight ROIs, majorly located in frontal and temporal lobes; (4) Conclusion: The present study proves an overall grey matter decline with normal ageing as reported previously. The provided reference may be used for detection of grey matter alterations in neurodegenerative diseases that are not apparent in standard MR scans, indicating the potential of using qMRI as an add-on diagnostic tool in a clinical setting.

## 1. Introduction

With increasing life expectancy world-wide, age-related neurodegenerative diseases pose a challenge to all healthcare systems, as currently, no preventive or curative treatment is available [1,2,3]. One prominent example of neurodegenerative diseases is amyotrophic lateral sclerosis (ALS), an adult-onset neurodegenerative disease characterized by rapidly progressive muscle weakness due to the degeneration of motor neurons, leading to progressive paralysis, breathing difficulties, swallowing problems and eventually early death [3]. However, symptomatic treatments may improve the patients’ quality of life [4,5] and disease-modifying agents such as Riluzole or Edaravone can slow the disease progression, if applied sufficiently early in the disease course [6,7]. Consequently, early diagnosis is essential to enable more efficient treatment. Since the diagnoses of these neurodegenerative diseases are commonly based on clinical criteria describing symptoms [1,2,3], early clinical diagnosis can be challenging, as in the early stages the symptoms can be unspecific and subtle [8]. Additional diagnostic tools, such as standard magnetic resonance imaging (MRI) scans, have not been able to increase diagnostic certainty in early stages of neurodegenerative diseases due to lack of specific findings [9], e.g., standard MRI scans are considered to be generally normal in patients with ALS (https://www.ninds.nih.gov/disorders/Patient-Caregiver-Education/Fact-Sheets/Amyotrophic-Lateral-Sclerosis-ALS-Fact-Sheet). In recent years, a number of MRI studies documented abnormal findings in patients with neurodegenerative disease using three-dimensional (3D) T2*-weighted sequences, but these signs are often nonspecific [10]. More sensitive MR imaging methods, such as quantitative MR measurements, have shown promise to bridge this gap through the identification of brain alterations in neurodegenerative diseases that are invisible in standard MRI scans. Previous studies with quantitative MR measurements found certain brain morphological differences such as the shrinking of the cortex and subcortical grey matter volume in Alzheimer’s disease and Parkinson’s disease patients [11,12], reduced cortical thickness spread from the primary motor cortex to extra-primary motor cortex in ALS patients, with reduced brain cortical thickness in primary motor cortex even being suggested as a hallmark of ALS progression [13,14,15,16,17,18]. Therefore, incorporating quantitative MR morphological measurements into clinical routine diagnostic use may help to recognize early alterations in individual patients with neurodegenerative disease. In our study, we established a reference database of regional cortical thickness (rCTh) and subcortical grey matter volumes (SGMV) from 120 healthy subjects which may be used to quantitatively estimate brain structural alterations beyond normal aging in patients with neurodegenerative diseases. We tested the accuracy of the classification based on the reference database in a small group of patients with amyotrophic lateral sclerosis.

## 2. Materials and Methods

### 2.1. Subjects

The human study was approved by the Ethics Committee of Hannover Medical School in the statement numbered 6167, and written informed consent was obtained from each participant. Healthy volunteers were recruited via an advertisement from the local community and rated by an experienced neurologist and a doctoral fellow together in our MR research center. Only those subjects were included who had no history of neurological disease, psychiatric disorder, or other major medical illness based on self-reports. To exclude cognitive or psychiatric impairment, two screening tests were performed by each subject (informed consent was obtained from the legal guardian where necessary): (1) The Beck Depression Inventory (BDI-II) test containing 21 multiple-choices questions with a score up to 8 for health condition; (2) The Dementia Detection test (DemTect test) containing five short tasks with a score from 13 to 18 for normal cognitive ability. Fifteen subjects were excluded due to abnormal BDI-II scores (≥9, *n* = 5), obesity (body mass index (BMI) > 30, *n* = 3) or underweight (BMI < 19, *n* = 1), artefacts in MR imaging (*n* = 1), abnormal MR findings (*n* = 2), chronic arterial hypertension (*n* = 1), and incomplete MR examinations (*n* = 2). Finally, 120 subjects aged between 21–70 years (mean age = 43.95 ± 15.03 years, BDI-II mean score = 2.32 ± 2.65, and DemTect mean score = 17.54 ± 0.99) were studied. To obtain an evenly distributed age range, there were at least 10 males and 10 females in each of the age decades, i.e., the 3rd decade (21–30 years), 4th decade (31–40 years), 5th decade (41–50 years), 6th decade (51–60 years), and 7th decade (61–70 years).

Patients suffering from ALS were recruited from the outpatient clinic at our Medical School. All ALS patients were diagnosed according to the revised El Escorial criteria [19], underwent Montreal cognitive assessment (MoCA) for cognitive function estimation [20], and received MR examinations as a part of the diagnostic procedures. Patients suffering from additional neurological disease, tumors, or with an age of more than 70 years (due to lack of matched healthy controls) were excluded. Finally, 11 patients (aged 39–59 years, mean age = 52.45 ± 6.80 years) diagnosed with ALS were included, with seven patients diagnosed with definite ALS, three with probable ALS, and one with possible ALS according to the revised El Escorial criteria.

### 2.2. Data Acquisition and Processing

All brain MR examinations were carried out under clinical routine conditions at the same 3.0 Tesla system (Verio, Siemens, Erlangen, Germany) with a 12-channel phased array head coil. The subjects were scanned among others with an axial T1-weighted Three-Dimensional Magnetization Prepared Rapid Gradient Echo (3D-MPRAGE) sequence (160 contiguous axial slices with an in-plane field of view 256 × 224 mm^2^ and a 1 mm isotropic voxel resolution TR/TE/TI = 1900/2.93/900 ms, flip angle 9°, acceleration factor = 2), with a scan time of 3.5 min. The data acquired with 3D-MPRAGE sequence were analyzed with the free software Computational Anatomy Toolbox (CAT12) (http://www.neuro.uni-jena.de/cat/), a toolbox attached to software package SPM 12 (http://www.fil.ion.ucl.ac.uk/spm/software/spm12). CAT12 was chosen for this study since previous studies reported that CAT12 was a more advanced and computationally efficient brain segmentation tool, providing a more accurate volumetric analysis [21,22,23].

Regarding its feasibility in clinical routine use, the image data analysis was done by automatically running the default processing pipelines without considering computationally expensive options provided in CAT12 [24,25]. The main procedures are briefly described as follows: the original MR data in Digital Imaging and Communications in Medicine (DICOM) format were transformed into Network Interface to File Transfer in the Internet (NIFTI) format, and underwent spatial normalization, brain extraction, segmentation of grey matter (GM), white matter (WM) and cerebrospinal fluid (CSF), and alignment to Montreal Neurological Institute standard space (MNI-152 template). Based on the projection-based thickness (PBT) method [26], the central surface, which lies between the grey matter/CSF boundary and the grey matter/white matter boundary, as well as the cortical thickness, defined as the distance between the grey/white boundary and the grey/CSF boundary, were estimated by use of surface-based morphometry (SBM) offered in CAT12 [26]. Subsequently, the values of rCTh (in mm) were extracted from 68 regions of interest (ROIs), with 34 ROIs in each hemisphere selected according to the Desikan-Killiany atlas (13 ROIs in the frontal lobe, 4 in the occipital lobe, 7 in parietal lobe, and 10 in temporal lobe) [27]. Here, we decided to measure rCTh rather than to measure regional cortical volume because it has been reported that rCTh was more sensitive to pathological changes [28]. In parallel, the estimated total intracranial volume (eTIV, in cm^3^) and the values of SGMV (determined in ratio to eTIV [29]) from seven ROIs in each hemisphere (accumbens area, amygdala, caudate, pallidum, putamen, hippocampus, and thalamus proper) were derived by use of voxel-based morphometry (VBM) contained in CAT12 [30]. Since all images revealed a bias index (≥77%) that was higher than the satisfactory level (75%) defined in CAT12 toolbox (http://www.neuro.uni-jena.de/cat/), no additional bias correction on the scans was performed.

### 2.3. Statistical Analysis

The normality of the data distribution was checked by using Shapiro-Wilk tests and quantile-quantile plots. For healthy subjects, two-sided *t*-test revealed gender differences only in estimated volumes of bilateral thalamus proper. Therefore, the values measured from these two ROIs were analyzed separately for males and females, while those from all other ROIs were combined. One-way ANOVA with trend analysis was carried out to estimate possible relationship between age and measured rCTh or SGMV values, which did not reveal a significant quadratic or higher order relationship except a linear relationship. Therefore, a linear regression analysis was used to estimate age dependences of the values of rCTh or SGMV, where a Bonferroni-corrected significance level alpha = 0.05/k was used, with k being the number of selected ROIs, i.e., alpha = 0.05/68 = 0.0007 for the analysis of rCTh values and alpha = 0.05/14 = 0.004 for the analysis of the SGMV values. Results with *p* < α were considered as statistically significant, and those with α < *p* < 0.05 as not statistically significant but showing a trend of linear correlation between age and measured values. In both cases, linear fits and calculations of 95% confidence interval bands and 95% prediction interval bands were carried out, where the 95% confidence interval bands demonstrated the goodness of the linear fits, while the 95% prediction interval bands demonstrated expected rCTh or SGMV values in healthy subjects of corresponding age range.

In ALS group alterations of rCTh or SGMV were estimated by taking the values measured in healthy subjects as normative reference database. In doing this, the rCTh and SGMV values of the patients were first qualitatively compared to those of healthy subjects by overlaying them graphically onto the normative reference database according to age, which allowed a direct visual observation of possible differences of brain rCTh or SGMV values between patients and healthy subjects. Shapiro-Wilk tests revealed non-normal distribution of the data measured from patients. Therefore, non-parametric Wilcoxon testing for two matched samples was used for comparison of brain rCTh and SGMV values between the patient group and matched healthy control group. The control group was composed of 11 subjects selected from the healthy subject collective, who matched to patients on a one-to-one basis for age (difference ≤ 2 years) and sex. The false discovery rate (FDR) suggested by Glickmann et al. [31] was used for the correction of multiple testing. The results with *p* < 0.05 that were confirmed by the FDR correction were considered as statistically significant, and those with *p* < 0.05 but that were not significant after the FDR correction were considered as showing a changing trend of observed alterations. All analyses were performed using SPSS software, version 24 and version 26 (IBM, Armonk, New York, NY, USA) and the graphics were drawn with the software OriginPro 2016G (OriginLab, Northampton, MA, USA).

## 3. Results

### 3.1. Healthy Ageing Human

The rCTh values and the SGMV values measured from all healthy subjects drawn against age are shown in Figure 1A,B, Appendix A (rCTh), and Figure 2 (SGMV). These figures also show the linear fits together with the 95% confidence interval bands and the 95% prediction interval bands for those ROIs revealing significant or trended age-related differences. Linear regression analysis revealed that with age, the rCTh values decreased significantly in 49 of 68 ROIs (21 ROIs in the frontal lobe, 12 in the temporal lobe, 10 in the parietal lobe, and 6 in the occipital lobe, *p* < 0.0007), and with a trend (0.0007 < *p* < 0.05) in 14 ROIs (4 ROIs each in frontal, temporal, and parietal lobes, respectively, and 2 ROIs in occipital lobe). The rCTh decreasing rate varied from −0.920% (in the left superior parietal) to −3.809% (in the right transverse temporal) per decade (Table 1). The SGMV values decreased with age significantly in 9 ROIs (bilateral accumbens area, caudate nucleus, hippocampus, putamen, and right amygdala, *p* < 0.004), and with a trend in the left amygdala and in the right thalamus of males (0.004 < *p* < 0.05). The decreasing rate of the SGMV varied from −2.038% (right amygdala) to −4.101% (left accumbens area) per decade (Table 2).

### 3.2. Patients with Amyotrophic Lateral Sclerosis

The characteristics of the patients are given in Table 3. All the patients revealed normal findings in standard brain MRI scans as commonly seen in ALS patients.

Examples of patients’ rCTh values (in 15 ROIs) and SGMV values (in 12 ROIs) are shown in Figure 3 and Figure 4, respectively, together with the normative reference data, which were represented either by the prediction interval bands or by the measured values of the healthy subjects. It is observable from Figure 3 and Figure 4 that in several ROIs the rCTh values of the patients were lower than or close to the lower boundary of the prediction interval bands of corresponding age ranges, while most of the SGMV values of the patients did not differ much from the reference data. Consistently, the quantitative Wilcoxon testing revealed that in comparison to the values of age- and sex-matched healthy controls selected from the reference data, the rCTh values of the ALS patients decreased (−6.06% to −10.18%) significantly (*p* < 0.05 and confirmed after FDR correction) in eight ROIs that were located in temporal lobes, and showed a decreasing trend (*p* < 0.05 but this was not confirmed after FDR correction) in seven ROIs located in frontal (four ROIs), temporal (two ROIs) and parietal (one ROI) regions (Table 4), while the SGMV values in ALS patients did not show significant or trended alterations.

## 4. Discussion

This study determined brain rCTh values in 68 ROIs and SGMV values in 14 ROIs from 120 healthy adults aged evenly distributed between 21 to 70 years by use of free software CAT12 and created a normative reference database showing age-related differences of rCTh and SGMV in healthy ageing. Moreover, MR data of 11 patients with amyotrophic lateral sclerosis were used to investigate the accuracy of the database to detect subtle grey matter alterations beyond normal ageing in patients with neurodegenerative disease.

One of our major observations was the age-related differences of rCTh and SGMV in healthy human brain, as reported previously [32,33]. We found that the values of rCTh in healthy ageing subjects were significantly different and decreased with age in a majority of measured ROIs (49/64), mostly within the frontal lobe (21/49), but also within the temporal lobe (12/49), the parietal lobe (10/49), and the occipital lobe (6/49) (Table 1), indicating a spatial distribution of age-related alteration in brain cortical thickness. Consistently, the rate of rCTh alteration (per decade) also varied among the brain regions, with a relatively strong negative correlation to age (the magnitude of Pearson’s coefficient |r| > 0.57) and a rate varying from −1.86% to −3.81% per decade in six ROIs (left precentral, left and right superior frontal, left postcentral, and left and right transverse temporal), an intermediate correlation to age (0.50 < |r| ≤ 0.57), a rate varying from −1.45% to −2.12% per decade in 8 ROIs (right pars triangularis, right precentral, right middle frontal, left middle temporal, left superior temporal, left lingual, right postcentral regions, and right fusiform), and a weak correlation to age (|r| ≤ 0.50) in the remaining 35 ROIs. Regarding SGMV, we found that the values of SGMVs were also significantly different in healthy ageing subjects and exhibited a negative linear correlation to age in 9 of 14 measured ROIs (bilateral accumbens area, caudate, hippocampus, putamen, and right amygdala) with |r| varying from 0.32 to 0.53 and a declining rate from −2.04% to −4.10% per decade (Table 2). Our observations were consistent with those reported previously [34,35,36]. For example, Zheng et al., in a study on 54 healthy adults aged 21–71 years [35] and Potvin et al., in two studies on more than 2700 healthy individuals aged 18–94 years [34,36] described age-related differences of rCTh and SGMV in healthy subjects in their studies. Although a different software (FreeSurfer https://surfer.nmr.mgh.harvard.edu/) was used to estimate rCTh and SGMV in these studies, their results were mostly similar to ours, proving the reliability of the present measurements. There are also some discrepancies to our study in detailed results: while all three studies found a linear decline of rCTh or SGMV with age in healthy ageing human brain, additional quadratic correlations of SGMV with age were reported by Zheng et al. [35], and additional quadratic or cubic correlations of rCTh and SGMV with age by Potvin et al. [34,36]. These discrepancies may be caused by differences in age distributions and sample sizes of studied subjects, as well as the heterogeneity of the used software and significance levels setting for data analysis in all three studies as also pointed by others: a sample size of 120 healthy subjects with an evenly age distribution between 21 to 70 years and a conservative Bonferroni-corrected significance level of 0.0007 were taken in present study, while a sample size of 54 with an age range between 21 to 71 years and a significance level of 0.001 were used by Zheng et al. [35], and a large sample size of 2713 subjects (age range between 18 to 94 years) pooled from 23 samples provided by 21 independent research groups with different age distributions and a significance level of 0.05 or 0.01 were used in the study of Potvin et al. [34,36]. According to Seiger et al. [25] the rCTh of insula produced by the CAT12 toolbox was almost 25% higher than values produced by Freesurfer; this could also be the reason for the variability. Additionally, it has to be mentioned that due to magnetic field distortions caused by neighboring structures containing bone or air, several ROIs showed relatively high variable rCTh values (e.g., temporal pole or insula, Appendix A) or SMGV values; therefore, the results derived from these ROIs (e.g., only male participants showed a trend of shrinking of the right, Figure 2) should be interpreted and used with caution.

It is noticeable that while the results of different studies, including ours, revealed a consistent age-related decline of brain cortical thickness and subcortical grey matter volume in healthy ageing human [34,35,36,37], the present study used the advanced and computationally efficient software CAT12 [21], and brain atlas defined ROIs to obtain rCTh and SGMV values [25], allowing the determination of rCTh or SGMV values in standardized brain regions of individual subjects. These properties may favor the method to be used in a clinical diagnostic setting to measure the brain morphological alterations in individual patients.

A further result of our study was that, using the values measured from our healthy subjects as normative reference data, altered rCTh was detected in multiple brain areas of the ALS patient group, who otherwise showed normal brain findings in standard MRI scans. While the graphical observations from Figure 3 and Figure 4 qualitatively indicated a decrease of the rCTh values in several brain regions of the patients, the Wilcoxon testing confirmed the findings: in comparison to age- and sex-matched healthy subjects, the ALS patients revealed decreases of rCTh significantly in eight temporal ROIs (Table 4), without significant changes in the SGMV values. Our observations of ALS-related decreases of temporal rCTh were consistent with those reported previously [13,14,15,16,17,18,38]: the thinning of temporal cortex regions was also reported by others and found to be related to bulbar involvement and associated with a rapidly progressive disease course [34]. Our result also showed a trend of precentral cortex thinning, which was considered as the hallmark of ALS pathology [12,13,14], though it did not show significant rCTh decrease after FDR correction. The possible reason could be the neurostructural heterogeneity and clinical symptom variation of ALS patients; in addition, the small sample of the patients could be a reason. We could, however, not confirm previous reported significant ALS-related decreases of SGMV [39,40,41]. This discrepancy could be explained by the fact that most of our ALS patients showed a nearly unimpaired cognitive performance with a mean MoCA score over 26 (Table 3) [20]; therefore, little alteration in cognition-related subcortical grey matter structures was to be expected, which is consistent with the results reported by Bede et al. [42], who found that ALS patients without cognitive or behavioral deficits did not reveal ALS-related volume reduction in any subcortical grey matter structure. Our results, derived from a small sample of ALS-patients, showed that, with the values measured from healthy ageing subjects as a normative reference database, it is possible to detect disease-related grey matter alterations in ALS patients invisible in standard brain MRI with quantitative MR morphologic measurements, demonstrating again its potential for clinical diagnostic use.

There were some limitations in this study: the sample size of 120 and the age range of 21 to 70 years for healthy subjects are relatively limited for an ageing study. Some factors, such as handedness or level of education of the healthy subjects, were not considered, which may impact the results. In the present study, we did not perform a detailed visual quality control of specific regional segmentation or additional bias correction because a higher bias index level indicated satisfactory quality of the imaging data, which, however, may generally allow the inclusion of poor-quality segmentations that influence the results of imaging studies. Another limitation that should be mentioned is the possible contamination of the elderly subjects with undiagnosed neurodegenerative conditions such as Alzheimer’s disease as a confounder, although this is unlikely to change the final results significantly, because all subjects revealed normal cognitive ability by DemTect testing. Due to small sample size of the patients, the influence of the medications as well as the possible correlations between brain changes and clinical symptoms could not be reliably analyzed. Further studies with a larger sample size are necessary to validate the results.

In conclusion, normal ageing in the healthy human brain is associated with an overall grey matter decline, which presented as rCTh thinning that predominantly affected the frontal lobe and SGMV shrinking. The results derived from healthy subjects may serve as a normative reference database for the clinical application of quantitative MR morphologic measurements to detect early affection of grey matter in neurodegenerative diseases.

## Figures and Tables

**Figure 1 brainsci-11-00055-f001:**
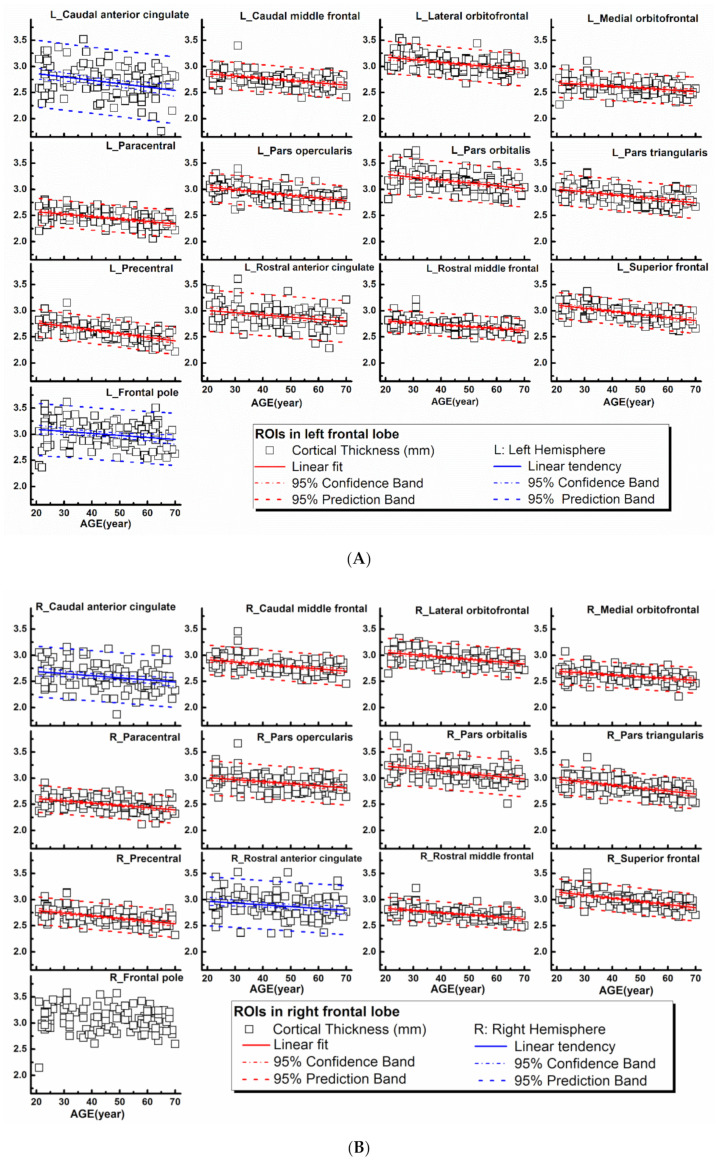
Regional cortical thicknesses (*y*-axis, in mm) measured from frontal regions of interest (ROIs) plotted versus age (*x*-axis). Corresponding linear fits, the calculated 95% confidence interval bands, and the 95% prediction interval bands were also shown in 25 ROIs, where a significant (*p* < 0.0007, 21 ROIs with red lines) or a trend of (0.0007 < *p* < 0.05, 4 ROIs with blue lines) correlation between the measured values and the age was found.

**Figure 2 brainsci-11-00055-f002:**
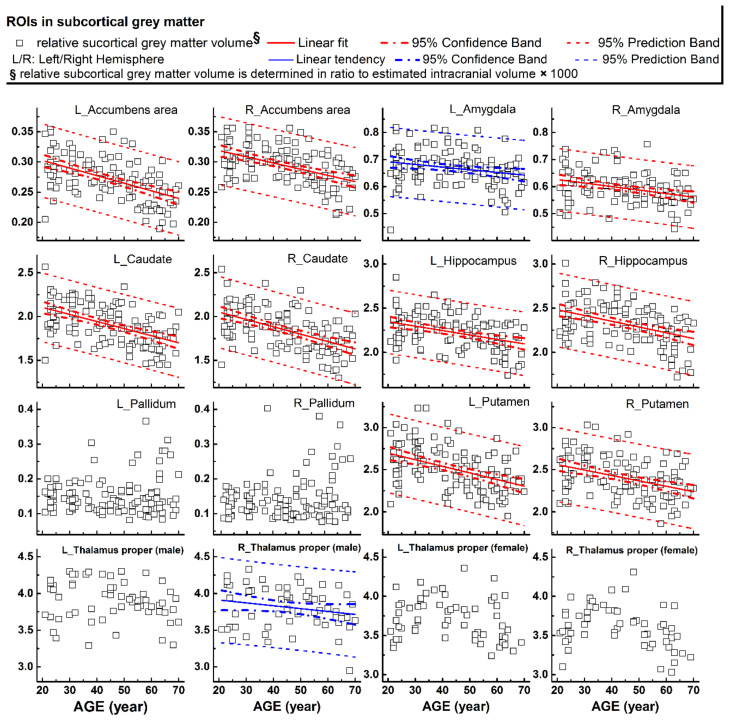
Relative volumes (*y*-axis) measured from subcortical grey matter structures plotted versus age (*x*-axis). Corresponding linear fits, the calculated 95% confidence interval bands, and the 95% prediction interval bands were also shown in 11 ROIs, where a significant (*p* < 0.004, 9 ROIs with red lines) or a trend of nearly significant (0.004 < *p* < 0.05, 2 ROIs with blue lines) correlation between the measured values and the age was found.

**Figure 3 brainsci-11-00055-f003:**
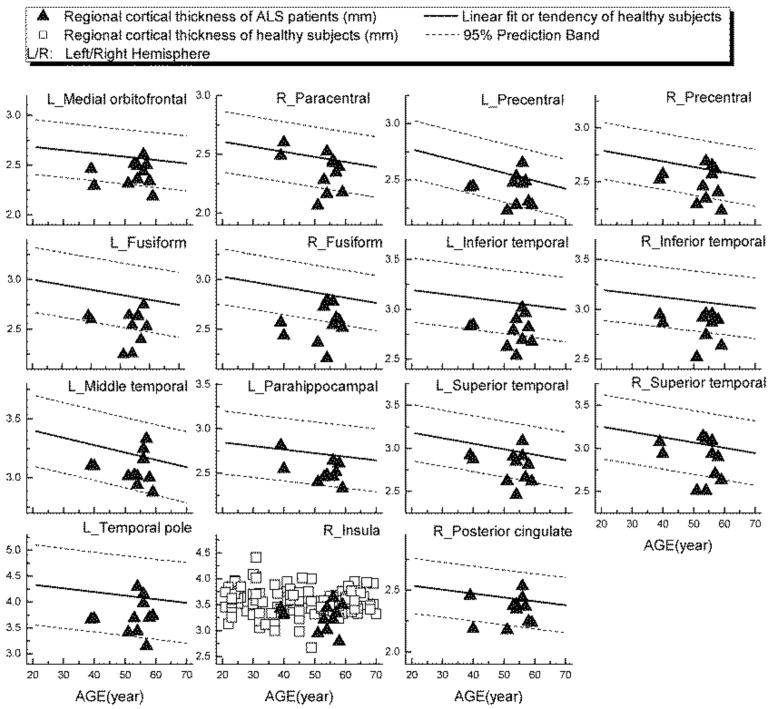
Regional cortical thickness (in mm) measured from ALS patients in 15 ROIs (triangular), together with the normative reference data, which were represented either by the prediction interval bands together with the linear fits or by the measured values of the healthy subjects.

**Figure 4 brainsci-11-00055-f004:**
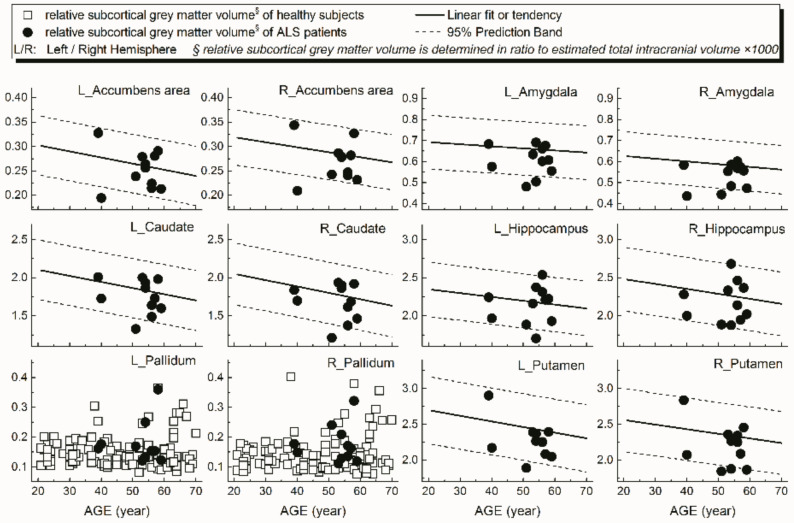
Relative volumes (*y*-axis) measured from subcortical grey matter structures of ALS patients in 12 ROIs (dot) together with the normative reference data, which were represented either by the prediction interval bands together with the linear fits or by the measured values of the healthy subjects.

**Table 1 brainsci-11-00055-t001:** Results of linear regression analyses of regional cortical thickness (rCTh) values by age in healthy subjects.

ROIs	Left Hemisphere (N = 120)							Right Hemisphere (N = 120)						
	r	*p*	Intercept		Slope		Variation ^α^	r	*p*	Intercept		Slope		Variation ^α^
			Mean	SE	Mean (×10^−2^)	SE (×10^−2^)	% Per Decade			Mean	SE	Mean (×10^−2^)	SE (×10^−2^)	% Per Decade
**Frontal**														
Caudal anterior cingulate	−0.292 *	0.0012	2.991	0.090	−0.643	0.194	−2.206	−0.242 *	0.0077	2.767	0.068	−0.399	0.147	−1.457
Caudal middle frontal	−0.448 **	0.0000	2.945	0.037	−0.432	0.080	−1.483	−0.428 **	0.0000	3.001	0.040	−0.438	0.085	−1.475
Lateral orbitofrontal	−0.441 **	0.0000	3.277	0.043	−0.497	0.093	−1.535	−0.444 **	0.0000	3.145	0.039	−0.448	0.083	−1.439
Medial orbitofrontal	−0.343 **	0.0001	2.751	0.039	−0.332	0.084	−1.213	−0.393 **	0.0000	2.761	0.035	−0.346	0.075	−1.261
Paracentral	−0.479 **	0.0000	2.664	0.037	−0.467	0.079	−1.783	−0.452 **	0.0000	2.694	0.036	−0.431	0.078	−1.622
Pars opercularis	−0.496 **	0.0000	3.147	0.039	−0.524	0.084	−1.691	−0.346 **	0.0001	3.090	0.046	−0.396	0.099	−1.291
Pars orbitalis	−0.417 **	0.0000	3.393	0.050	−0.542	0.109	−1.620	−0.398 **	0.0000	3.328	0.048	−0.492	0.104	−1.495
Pars triangularis	−0.464 **	0.0000	3.106	0.042	−0.521	0.091	−1.704	−0.516 **	0.0000	3.091	0.040	−0.566	0.086	−1.866
Precentral	−0.631 **	0.0000	2.912	0.037	−0.700	0.079	−2.481	−0.505 **	0.0000	2.892	0.037	−0.507	0.080	−1.784
Rostral anterior cingulate	−0.311 **	0.0005	3.091	0.056	−0.430	0.121	−1.404	−0.21 6*	0.0176	3.035	0.066	−0.343	0.142	−1.134
Rostral middle frontal	−0.448 **	0.0000	2.878	0.031	−0.363	0.067	−1.270	−0.510 **	0.0000	2.915	0.030	−0.417	0.065	−1.445
Superior frontal	−0.583 **	0.0000	3.223	0.035	−0.588	0.075	−1.859	−0.576 **	0.0000	3.263	0.036	−0.598	0.078	−1.868
Frontal pole	−0.228 *	0.0123	3.169	0.070	−0.385	0.151	−1.222	−0.157	0.0856					
**Occipital**														
Cuneus	−0.377 **	0.0000	2.213	0.037	−0.352	0.080	−1.613	−0.319 **	0.0004	2.191	0.042	−0.328	0.090	−1.515
Lateral occipital	−0.265 *	0.0035	2.462	0.041	−0.266	0.089	−1.083	−0.205 *	0.0250	2.378	0.043	−0.210	0.093	−0.882
Lingual	−0.545 **	0.0000	2.359	0.032	−0.488	0.069	−2.120	−0.459 **	0.0000	2.296	0.034	−0.414	0.074	−1.837
Pericalcarine	−0.488 **	0.0000	2.048	0.040	−0.526	0.087	−2.661	−0.397 **	0.0000	2.017	0.041	−0.412	0.088	−2.091
**Parietal**														
Inferior parietal	−0.309 **	0.0006	2.771	0.035	−0.266	0.075	−0.960	−0.294 *	0.0011	2.761	0.037	−0.265	0.079	−0.960
Isthmus cingulate	−0.233 *	0.0103	2.572	0.061	−0.344	0.132	−1.348	−0.224 *	0.0138	2.721	0.056	−0.300	0.120	−1.106
Postcentral	−0.594 **	0.0000	2.504	0.033	−0.573	0.071	−2.356	−0.518 **	0.0000	2.589	0.034	−0.489	0.074	−1.927
Posterior cingulate	−0.409 **	0.0000	2.658	0.040	−0.415	0.085	−1.582	−0.393 **	0.0000	2.599	0.032	−0.316	0.068	−1.223
Precuneus	−0.279 *	0.0020	2.654	0.039	−0.262	0.083	−0.988	−0.324 **	0.0003	2.654	0.034	−0.275	0.074	−1.038
Superior parietal	−0.328 **	0.0003	2.445	0.028	−0.225	0.060	−0.920	−0.332 **	0.0002	2.493	0.031	−0.254	0.067	−1.020
Supramarginal	−0.462 **	0.0000	2.856	0.038	−0.458	0.081	−1.626	−0.374 **	0.0000	2.898	0.038	−0.355	0.081	−1.232
**Temporal**														
Banks sts	−0.317 **	0.0004	2.905	0.048	−0.378	0.104	−1.311	−0.166	0.0702					
Entorhinal	−0.305 *	0.0007	4.462	0.120	−0.901	0.259	−2.066	−0.203 *	0.0259	4.231	0.134	−0.651	0.288	−1.558
Fusiform	−0.425 **	0.0000	3.098	0.046	−0.504	0.099	−1.651	−0.500 **	0.0000	3.133	0.039	−0.528	0.084	−1.712
Inferior temporal	−0.346 **	0.0001	3.272	0.045	−0.392	0.098	−1.204	−0.344 **	0.0001	3.269	0.043	−0.368	0.092	−1.130
Middle temporal	−0.532 **	0.0000	3.526	0.042	−0.623	0.091	−1.798	−0.476 **	0.0000	3.492	0.043	−0.542	0.092	−1.572
Parahippocampal	−0.325 **	0.0003	2.927	0.050	−0.403	0.108	−1.390	−0.234 *	0.0100	2.880	0.064	−0.360	0.138	−1.258
Superior temporal	−0.511 **	0.0000	3.309	0.046	−0.639	0.099	−1.972	−0.447 **	0.0000	3.374	0.053	−0.615	0.113	−1.857
Temporal pole	−0.264 *	0.0035	4.474	0.109	−0.702	0.236	−1.590	−0.027	0.7686					
Transverse temporal	−0.609 **	0.0000	2.840	0.055	−0.992	0.119	−3.694	−0.615 **	0.0000	2.931	0.058	−1.053	0.124	−3.809
Insula	−0.055	0.5500						−0.125	0.1727					

ROIs: regions of interest, defined according to the Desikan-Killiany atlas (Desikan et al., 2006). r: Pearson’s coefficient. SE: Standard Error. ** showing significant linear correlation (*p* < 0.0007, Bonferroni corrected significance level). * Not significant but showing a tendency of age dependence (0.0007 < *p* < 0.05). ^α^ Ratio of the values at age 21.

**Table 2 brainsci-11-00055-t002:** Results of linear regression of relative SGMV values to age in healthy subjects.

ROIs	N	r	*p*	Intercept		Slope		Variation ^α^
				Mean	SE	Mean (×10^−2^)	SE (×10^−2^)	% per decade
L_Accumbens area	120	−0.531 **	0.0000	0.328	0.009	−0.126	0.018	−4.101
R_Accumbens area	120	−0.484 **	0.0000	0.340	0.008	−0.103	0.017	−3.176
L_Amygdala	120	−0.227 *	0.0126	0.712	0.018	−0.099	0.039	−1.397
R_Amygdala	120	−0.324 **	0.0003	0.652	0.016	−0.130	0.035	−2.038
L_Caudate	120	−0.530 **	0.0000	2.267	0.055	−0.809	0.119	−3.780
R_Caudate	120	−0.530 **	0.0000	2.220	0.057	−0.840	0.124	−4.029
L_Hippocampus	120	−0.390 **	0.0000	2.447	0.051	−0.503	0.109	−2.105
R_Hippocampus	120	−0.427 **	0.0000	2.613	0.059	−0.653	0.127	−2.585
L_Pallidum	120	0.092	0.3154					
R_Pallidum	120	0.151	0.0991					
L_Putamen	120	−0.452 **	0.0000	2.854	0.066	−0.783	0.142	−2.853
R_Putamen	120	−0.405 **	0.0000	2.689	0.062	−0.643	0.134	−2.467
Female_L_thalamus proper	61	−0.198	0.1267					
Female_R_thalamus proper	61	−0.210	0.1046					
Male_L_thalamus proper	59	−0.184	0.1632					
Male_R_thalamus proper	59	−0.321 *	0.0131	3.862	0.102	−0.576	0.225	−1.509

ROIs: regions of interest, defined according to Neuromorphometrics atlas (Strudwick Caviness et al., 1999). r: Pearson’s coefficient. SE: Standard Error. L/R: left/right hemisphere. ** showing significant linear correlation (*p* < 0.004, Bonferroni corrected significant level). * Not significant but showing a tendency of age dependence (0.004< *p* < 0.05). ^α^ Ratio of the values at age 21.

**Table 3 brainsci-11-00055-t003:** Characteristics of the ALS patients and age- and sex-matched health controls.

ALS Patients								Age- and Sex-Matched Health Controls		
Number	Sex	Age (Years)	Time between First Symptom and MRI (Months)	MoCA	Onset	Medication	Diagnosis According to Revised El Escorial Criteria	Sex	Age (Years)	DemTect
1	female	58	5	28	spinal	No medication	Probable ALS	female	59	14
2	male	39	6.5	28	spinal	No medication	Probable ALS	male	40	18
3	male	57	51	29	spinal	No medication	Probable ALS	male	55	18
4	female	56	16	30	spinal	Cymbalta, Tamsulosin	Definite ALS	female	57	18
5	female	40	22	30	spinal	Citalopram, L-Thyroxin	Definite ALS	female	42	18
6	male	51	Unknown	Unknown	spinal	Unknown	Definite ALS	male	52	18
7	male	54	6	22	spinal	No medication	Definite ALS	male	52	17
8	male	56	14	28	spinal	Metformin, Enalapril, Bisoprolol	Definite ALS	male	58	17
9	male	54	Unknown	Unknown	bulbar	Unknown	Definite ALS (with pseudobulbar paralysis)	male	54	18
10	male	53	18	23	spinal	No medication	Possible ALS	male	52	18
11	male	59	Unknown	Unknown	spinal	Unknown	Definite ALS (with motor neuron disease)	male	60	18

MoCA: Montreal Cognitive Assessment. DemTect: Dementia Detection test. Revised El Escorial criteria (Brooks et al., 2000).

**Table 4 brainsci-11-00055-t004:** Wilcoxon tests of the rCTh values as well as the SGMV values measured from ALS patients and age- and sex-matched health controls.

ROIs	Left Hemisphere				Right Hemisphere			
	ALS Patients (*n* = 11)	Age- and Sex-Matched HCs (*n* = 11)	*p* (Wilcoxon Test)	Rel. Diff (%) ^#^	ALS Patients (*n* = 11)	Age- and Sex-Matched HCs (*n* = 11)	*p* (Wilcoxon Test)	Rel. Diff (%) ^#^
**rCTh (mm)**								
Frontal								
Caudal anterior cingulate	2.40 ± 0.39	2.61 ± 0.22	0.175		2.34 ± 0.26	2.52 ± 0.15	0.067	
Caudal middle frontal	2.62 ± 0.13	2.64 ± 0.12	0.331		2.71 ± 0.16	2.67 ± 0.10	0.520	
Lateral orbitofrontal	2.88 ± 0.14	2.95 ± 0.15	0.278		2.81 ± 0.11	2.88 ± 0.10	0.123	
Medial orbitofrontal	2.41 ± 0.12	2.56 ± 0.10	0.007 *	−5.859	2.43 ± 0.14	2.55 ± 0.08	0.083	
Paracentral	2.33 ± 0.19	2.41 ± 0.23	0.123		2.36 ± 0.17	2.48 ± 0.14	0.019 *	−4.839
Pars opercularis	2.73 ± 0.11	2.78 ± 0.08	0.206		2.75 ± 0.13	2.83 ± 0.13	0.147	
Pars orbitalis	3.01 ± 0.16	3.05 ± 0.14	0.638		2.95 ± 0.15	3.03 ± 0.10	0.240	
Pars triangularis	2.72 ± 0.12	2.76 ± 0.15	0.520		2.68 ± 0.10	2.73 ± 0.15	0.831	
Precentral	2.42 ± 0.13	2.51 ± 0.11	0.032 *	−3.054	2.49 ± 0.15	2.60 ± 0.08	0.042 *	−4.231
Rostral anterior cingulate	2.66 ± 0.20	2.77 ± 0.22	0.240		2.68 ± 0.27	2.79 ± 0.19	0.577	
Rostral middle frontal	2.59 ± 0.10	2.63 ± 0.08	0.365		2.61 ± 0.11	2.65 ± 0.06	0.413	
Superior frontal	2.77 ± 0.15	2.85 ± 0.11	0.102		2.82 ± 0.17	2.89 ± 0.12	0.240	
Frontal pole	2.91 ± 0.25	2.88 ± 0.19	0.700		2.98 ± 0.21	3.06 ± 0.21	0.520	
Occipital								
Cuneus	1.89 ± 0.13	1.97 ± 0.15	0.123		1.89 ± 0.13	1.96 ± 0.16	0.465	
Lateral occipital	2.21 ± 0.14	2.28 ± 0.09	0.278		2.15 ± 0.13	2.24 ± 0.09	0.175	
Lingual	1.97 ± 0.11	2.07 ± 0.10	0.123		1.97 ± 0.17	2.07 ± 0.09	0.320	
Pericalcarine	1.66 ± 0.11	1.72 ± 0.17	0.365		1.68 ± 0.14	1.78 ± 0.16	0.147	
Parietal								
Inferior parietal	2.56 ± 0.15	2.55 ± 0.10	1.000		2.51 ± 0.13	2.54 ± 0.09	0.311	
Isthmus cingulate	2.26 ± 0.19	2.30 ± 0.21	0.898		2.10 ± 0.23	2.47 ± 0.20	0.638	
Postcentral	2.12 ± 0.14	2.16 ± 0.08	0.365		2.21 ± 0.13	2.29 ± 0.09	0.175	
Posterior cingulate	2.30 ± 0.14	2.42 ± 0.17	0.102		2.34 ± 0.12	2.43 ± 0.13	0.042 *	−3.704
Precuneus	2.35 ± 0.15	2.44 ± 0.11	0.147		2.31 ± 0.20	2.43 ± 0.13	0.102	
Superior parietal	2.26 ± 0.14	2.26 ± 0.09	0.966		2.26 ± 0.14	2.31 ± 0.09	0.206	
Supramarginal	2.51 ± 0.13	2.52 ± 0.08	0.966		2.62 ± 0.13	2.62 ± 0.12	0.898	
Temporal								
Banks sts	2.60 ± 0.13	2.62 ± 0.13	0.520		2.63 ± 0.14	2.66 ± 0.15	1.000	
Entorhinal	4.04 ± 0.36	4.10 ± 0.25	0.966		3.89 ± 0.44	4.08 ± 0.22	0.520	
Fusiform	2.53 ± 0.16	2.76 ± 0.12	0.001 **	−8.333	2.56 ± 0.18	2.85 ± 0.14	0.002 **	−10.175
Inferior temporal	2.79 ± 0.15	3.00 ± 0.11	0.007 **	−7.000	2.84 ± 0.14	3.04 ± 0.15	0.003 **	−6.579
Middle temporal	3.07 ± 0.13	3.16 ± 0.13	0.019 **	−2.848	3.11 ± 0.21	3.18 ± 0.12	0.175	
Parahippocampal	2.52 ± 0.13	2.69 ± 0.20	0.019 **	−6.320	2.54 ± 0.17	2.64 ± 0.17	0.240	
Superior temporal	2.79 ± 0.18	2.97 ± 0.15	0.014 **	−6.061	2.87 ± 0.24	3.07 ± 0.13	0.042 *	−6.515
Temporal pole	3.72 ± 0.33	4.14 ± 0.36	0.032 *	−10.145	3.91 ± 0.41	4.20 ± 0.37	0.175	
Transverse temporal	2.11 ± 0.22	2.27 ± 0.21	0.067		2.23 ± 0.22	2.32 ± 0.20	0.365	
Insula	3.23 ± 0.20	3.43 ± 0.10	0.054		3.26 ± 0.26	3.55 ± 0.15	0.014 **	−8.169
**relative SGMV ^§^**								
Accumbens	0.25 ± 0.04	0.27 ± 0.03	0.413		0.27 ± 0.04	0.29 ± 0.02	0.638	
Amygdala	0.61 ± 0.07	0.67 ± 0.05	0.054		0.53 ± 0.06	0.60 ± 0.05	0.102	
Caudate	1.75 ± 0.23	1.76 ± 0.17	0.966		1.68 ± 0.24	1.72 ± 0.18	0.831	
Hippocampus	2.14 ± 0.25	2.23 ± 0.15	0.520		2.18 ± 0.26	2.30 ± 0.19	0.638	
Pallidum	0.18 ± 0.07	0.14 ± 0.03	0.320		0.18 ± 0.06	0.13 ± 0.04	0.147	
Putamen	2.23 ± 0.26	2.47 ± 0.20	0.083		2.21 ± 0.30	2.33 ± 0.16	0.240	
Thalamus proper	3.50 ± 0.45	3.71 ± 0.32	0.240		3.32 ± 0.53	3.58 ± 0.31	0.240	

HCs: health controls. ROI: Region of interest. ^§^ relative SGMV is determined in a ratio to the estimated total intracranial volume × 1000. ** Significant results after multiple comparisons correction by using false-discovery rate (FDR). * *p* < 0.05, but results are not significant after FDR correction. ^#^ Rel. Diff.: relative difference = (mean (patients)—mean (controls))/mean (controls) × %.

## Data Availability

The data presented in this study are storage at the Institute of Diagnostic and Interventional Neuroradiology, Hannover Medical School, and available on request for scientific research.

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
