# Peer review of "Clinically Applicable Quantitative Magnetic Resonance Morphologic Measurements of Grey Matter Changes in the Human Brain"

_brainsci, 2021, doi:10.3390/brainsci11010055_

Round 1
Reviewer 1 Report
Can you discuss in detail why specifically these 8 cortical areas showed significant thinning in ALS patients?
How do you explain that only Right hemisphere in only male participants showed a trend of shrinking of the Thalamus proper?
Fig. 3 shows the Left Posterior cingulate while Table 4 shows a decreasing trend for the Right Posterior cingulate (p=0.042) while for the Left Posterior cingulate p=0.102. What do you want to show in Fig. 2–Left or Right hemisphere of Posterior cingulate?
Minor:
In Table 1 & 2 description of R is missing.
Keep the correlation notation consistent in Tables 1 & 2 and in the Discussion- either small r or capital R.
In row 266 the r value has to be greater than or equal to 0.50.
Suggestion: The first part of paragraph 2 in Discussion that describes your finding for healthy participants better fits in the Results.
Author Response
Comments and Suggestions for Authors
Can you discuss in detail why specifically these 8 cortical areas showed significant thinning in ALS patients?
Answer: Thank you for the advice, we have modified the discussion correspondingly (lines 309-310).
How do you explain that only Right hemisphere in only male participants showed a trend of shrinking of the Thalamus proper?
Answer: We have now modified the text correspondingly (line 292).
Fig. 3 shows the Left Posterior cingulate while Table 4 shows a decreasing trend for the Right Posterior cingulate (p=0.042) while for the Left Posterior cingulate p=0.102. What do you want to show in Fig. 2–Left or Right hemisphere of Posterior cingulate?
Answer: Right Posterior cingulate was wrongly labeled as Left posterior cingulate. I am sorry for making this mistake. It has been now corrected in the updated manuscript Fig.3.
Minor:
In Table 1 & 2 description of R is missing.
Answer: As suggested we have now corrected it in Table 1 &2 and marked it with yellow shadow.
Keep the correlation notation consistent in Tables 1 & 2 and in the Discussion- either small r or capital R.
Answer: Thank you for pointing this inconsistent writing, we have made the changes to be a small r which is shadowed in yellow.
In row 266 the r value has to be greater than or equal to 0.50.
Answer: We have now revised it as suggested (line 274).
Suggestion: The first part of paragraph 2 in Discussion that describes your finding for healthy participants better fits in the Results.
Answer: Thank you for this suggestion. Since our purpose was to check whether the values derived from healthy subjects could be used as normative reference data, we would like to keep the description here to give the reader a overlook about the data, and as a contract to those concerning patients’ data.
Reviewer 2 Report
This study aimed to create a reference database of regional cortical thickness (rCTh) and subcortical grey matter volumes (SGMV) from a quite wide cohort of healthy subjects and to test it in a small group of patients with Amyotrophic Lateral Sclerosis. The age range of healthy subjects is wide, genders are equally (or quite equally) distributed in each decade, and the test on patients was done choosing data from healthy subjects matched for gender and age. The article is well-written and the method employed for data analysis is described clearly.
However, in my opinion there are some points that should be addressed.
- In the introduction (lines 45-47), the authors wrote that standard MRI is usually normal in the early stage of neurodegenerative diseases (note that the web page mentioned at line 48 is unavailable). That could be true if we consider a standard MRI as done just by T1-w and T2-w sequences, but the use of T2*-w sequences (especially 3D T2*-w sequences) changes the scenario. During the last years, indeed, a number of MRI studies documented abnormal findings in the substantia nigra of patients with Parkinson disease and in the primary motor cortex of patients with amyotrophic lateral sclerosis using 3D T2*-weighted sequences (and in patients with amyotrophic lateral sclerosis sometimes also using 2D GRE or T2-weighted sequences).
- The authors wrote that the relatively high variable values of rCTh and SGMV measures in some brain regions of healthy subjects (for example temporal pole, insula and thalamus) is due to magnetic field distortion caused by neighbouring structure containing bone or air (lines 290-293). It could be true for the temporal pole which is close to the mastoid and the sphenoidal sinus, but it seems improbable in insula or thalamus that are quite far from both bone and air. Which could be the cause of the variability in those brain regions?
- About the strength of Pearson correlation: to the best of my knowledge, the correlation is strong for Pearson’s coefficient (r) higher than 0.7 and moderate when it ranges between 0.5 and 0.7. In the discussion, however, the authors wrote “strong correlation” referring to r > 0.57 and intermediate correlation for 0.5 < r < 0.7. Can the author explain which reference has been used?
- A visual check of the segmentation was not done (line 324), but it would be useful to confirm the good quality of the ROI used in the study.
Author Response
Comments and Suggestions for Authors
This study aimed to create a reference database of regional cortical thickness (rCTh) and subcortical grey matter volumes (SGMV) from a quite wide cohort of healthy subjects and to test it in a small group of patients with Amyotrophic Lateral Sclerosis. The age range of healthy subjects is wide, genders are equally (or quite equally) distributed in each decade, and the test on patients was done choosing data from healthy subjects matched for gender and age. The article is well-written and the method employed for data analysis is described clearly.
However, in my opinion there are some points that should be addressed.
- In the introduction (lines 45-47), the authors wrote that standard MRI is usually normal in the early stage of neurodegenerative diseases (note that the web page mentioned at line 48 is unavailable). That could be true if we consider a standard MRI as done just by T1-w and T2-w sequences, but the use of T2*-w sequences (especially 3D T2*-w sequences) changes the scenario. During the last years, indeed, a number of MRI studies documented abnormal findings in the substantia nigra of patients with Parkinson disease and in the primary motor cortex of patients with amyotrophic lateral sclerosis using 3D T2*-weighted sequences (and in patients with amyotrophic lateral sclerosis sometimes also using 2D GRE or T2-weighted sequences).
Answer: Thank you for these valuable information and suggestion. The main trend utility of standard MRI is still the T1-w, T2-2 and FLAIR sequences. In some ALS patients, high signal intensity is evident along the corticospinal tract on T2 weighted, FLAIR and proton density MR images, but these signs are often nonspecific for ALS from patients (Turner MR, Kiernan MC, Leigh PN, et al. Biomarkers in amyotrophic lateral sclerosis. Lancet Neurol 2009;8:94e109.). We have now included this point in the introduction (Lines 51-53).
- The authors wrote that the relatively high variable values of rCTh and SGMV measures in some brain regions of healthy subjects (for example temporal pole, insula and thalamus) is due to magnetic field distortion caused by neighbouring structure containing bone or air (lines 290-293). It could be true for the temporal pole which is close to the mastoid and the sphenoidal sinus, but it seems improbable in insula or thalamus that are quite far from both bone and air. Which could be the cause of the variability in those brain regions?
Answer: The variability could also be caused by sample size or different applied software. Due to sex difference of the volume of bilateral thalamus proper was found in healthy subjects; it leads to smaller sample size in regression analysis according to sex, which may cause the variability. According to Seiger et al (Seiger et al. 2018), rCTh of insula produced by CAT 12 toolbox showed almost 25% higher than that produced by Freesurfer, this could also be the reason of variability.
We have modified this part to make the points to be more clear (Lines 287-288)
- About the strength of Pearson correlation: to the best of my knowledge, the correlation is strong for Pearson’s coefficient (r) higher than 0.7 and moderate when it ranges between 0.5 and 0.7. In the discussion, however, the authors wrote “strong correlation” referring to r > 0.57 and intermediate correlation for 0.5 < r < 0.7. Can the author explain which reference has been used?
Answer: Thanks for pointing it. We have now modified “strong” to relative strong” to differentiate it from the general definition, i.e. our considered “relatively strong correlation” referring to r > 0.57 and intermediate correlation for 0.5 < r < 0.57. (line 265).
- A visual check of the segmentation was not done (line 324), but it would be useful to confirm the good quality of the ROI used in the study.
Answer: We totally agree with you and mentioned it as a limitation of this study in the manuscript.
Reviewer 3 Report
The manuscript by Fu and colleagues describes an MRI study where a normative reference database shows age-related differences of cortical thickness and subcortical gray matter volume by using 120 healthy subjects, confirming a previously documented phenomenon. By using a very small cohort of patients with ALS (only 11 subjects), the Authors claim that “it is possible to detect disease-related grey matter alterations in ALS patients invisible in standard brain MRI with quantitative MR morphologic measurements, demonstrating its potential for clinical diagnostic use.”
The presented results concerning age-dependance of cortical thickness and subcortical gray matter volume largely confirm previous observations reported in the literature by several equipes. As acknowledged by the Authors, this part has some limitations, but they are honeslty discussed. In addition to what the Authors already did, a thorough visual quality control of specific regional segmentation could be performed and re-run the analysis only on verified good data, to achieve two goals: one is to verify that there is no change with respect to the results obtained with the entire cohort of healthy subjects; the second motivation would be to assess the success rate of the automated procedure to segment the regions of interest (it surely does not work equally well in all ROIs). This would add value to the manuscript.
The second part, concerning ALS patients, would be the most novel part of the study, but its major limitation is obviously the very small number of ALS patients (only 11). Of course I am aware that, being ALS a rare disease, this number cannot be increased in a revised version of this paper. The paper could however be re-structured to give an even smaller relative weight to the conclusions drawn from such a small patient cohort, and possibly give more weight to the confirmatory results about cortical thickness and subcortical gray matter volume changes detected with this pipeline in healthy ageing. In fact, by looking at Figures 3 and 4, one might argue that at this stage it is impossible to identify patients with ALS on the basis of their cortical thickness or subcortical gray matter volumes: the vast majority of data points of patients fall within the prediction band described by healthy subjects, and most patient data points outside the prediction band look anyways very similar to the data points of some healthy outliers. How can the Authors support the conclusion that these MR morphologic measurements could “detect early affection of grey matter in neurodegenerative diseases” and "have potential for diagnostic use"? Rather, my understanding is that these results at the current stage demonstrate that this type of information cannot be applied for clinical / diagnostic purpose. However, the use of more advanced techniques (for example machine learning?) might enable improvements in the future.
The mini-review paragraph of previous studies investigating morphological differences in the cortex of patients with ALS (lines 52-56) may include a few more papers, for example: Verstraete et al. (J Neurol Neurosurg Psychiatry 2012; doi:10.1136/jnnp-2011-300909 – this is cited only in the discussion); Walhout et al. (J Neurol Neurosurg Psychiatry 2015; doi:10.1136/jnnp-2013-306839); Donatelli et al. (American Journal of Neuroradiology 2018; DOI: https://doi.org/10.3174/ajnr.A5423).
While the primary motor cortex shows statistically significant differences between ALS patients and healthy controls, such differences do not reach the threshold of statistical significance when FDR correction is applied (Table 4). This observation probably deserves a bit of further discussion.
In the comparison between ALS patients and healthy controls, I find that the choice of discarding 109 healthy controls and use only the 11 subjects whose age best matches that of patients is not very wise. To enable paired, age-matched comparison between patients and controls, the authors could devise a way to use synthetic normative data extracted from the fit. For example, for comparison with patient number 1, aged 58, authors could use “synthetic data” generated from their fit, taken at age = 58. This would enable to: (1) prevent the risk that the matched control subject is actually an outlier; (2) avoid to thrash the wealth of the information available from the 120 controls.
Finally, I have also the following relative minor comments:
The title is very non-informative. It doesn’t mention ALS at all. Further, the “purpose” section of the abstract does not make any mention of ALS, either. This is inconsistent with the current design of the paper, whose introduction (in the main text) begins as if the paper is principally about ALS.
In the methods, it is fine to motivate the choice for CAT12, but the details about the unused tools perhaps could be shortened.
In figures 1 A&B: I suggest to re-draw the plots of cortical thickness and keep the y-axis fixed (for example between 1.5 and 4) to visually depict the relative difference between the slopes.
A few typos are present throughout the paper, but these minor problems could be addressed first by the authors themselves during revision, and/or at a later stage of the review process.
Author Response
Comments and Suggestions for Authors
The manuscript by Fu and colleagues describes an MRI study where a normative reference database shows age-related differences of cortical thickness and subcortical gray matter volume by using 120 healthy subjects, confirming a previously documented phenomenon. By using a very small cohort of patients with ALS (only 11 subjects), the Authors claim that “it is possible to detect disease-related grey matter alterations in ALS patients invisible in standard brain MRI with quantitative MR morphologic measurements, demonstrating its potential for clinical diagnostic use.”
The presented results concerning age-dependance of cortical thickness and subcortical gray matter volume largely confirm previous observations reported in the literature by several equipes. As acknowledged by the Authors, this part has some limitations, but they are honeslty discussed. In addition to what the Authors already did, a thorough visual quality control of specific regional segmentation could be performed and re-run the analysis only on verified good data, to achieve two goals: one is to verify that there is no change with respect to the results obtained with the entire cohort of healthy subjects; the second motivation would be to assess the success rate of the automated procedure to segment the regions of interest (it surely does not work equally well in all ROIs). This would add value to the manuscript.
Answer: Thank you very much for the valuable suggestions. Indeed, the total project is under further working and we will including what suggested by the reviewer. Here we would like to report the first part of the work. We agree that a visual check of segmentation would be useful to confirm the good quality of the ROI. We realize that this is a limitation of our study, and mentioned them accordingly in the manuscript.
The second part, concerning ALS patients, would be the most novel part of the study, but its major limitation is obviously the very small number of ALS patients (only 11). Of course I am aware that, being ALS a rare disease, this number cannot be increased in a revised version of this paper. The paper could however be re-structured to give an even smaller relative weight to the conclusions drawn from such a small patient cohort, and possibly give more weight to the confirmatory results about cortical thickness and subcortical gray matter volume changes detected with this pipeline in healthy ageing. In fact, by looking at Figures 3 and 4, one might argue that at this stage it is impossible to identify patients with ALS on the basis of their cortical thickness or subcortical gray matter volumes: the vast majority of data points of patients fall within the prediction band described by healthy subjects, and most patient data points outside the prediction band look anyways very similar to the data points of some healthy outliers. How can the Authors support the conclusion that these MR morphologic measurements could “detect early affection of grey matter in neurodegenerative diseases” and "have potential for diagnostic use"? Rather, my understanding is that these results at the current stage demonstrate that this type of information cannot be applied for clinical / diagnostic purpose. However, the use of more advanced techniques (for example machine learning?) might enable improvements in the future.
Answer:
Thanks for the suggestion. Indeed it could be an alternative just to write the manuscript with focus only the data from healthy subjects. However, as a research group located in a clinical unite, we aim to established MRI methods that could be used for diagnostic purpose. For this study all the MRI scans were carried out under clinical routine circumstance. We know that we have only a small sample of ALS patients. Anyhow, we could show in the discussion part that our results derived from the patients were mostly consistent with those reported by others. Therefore we would like to report these preliminary results in thei manuscript, while the project is undergoing. And in the study we will also consider more advanced techniques like machine learning.
The mini-review paragraph of previous studies investigating morphological differences in the cortex of patients with ALS (lines 52-56) may include a few more papers, for example: Verstraete et al. (J Neurol Neurosurg Psychiatry 2012; doi:10.1136/jnnp-2011-300909 – this is cited only in the discussion); Walhout et al. (J Neurol Neurosurg Psychiatry 2015; doi:10.1136/jnnp-2013-306839); Donatelli et al. (American Journal of Neuroradiology 2018; DOI: https://doi.org/10.3174/ajnr.A5423).
Answer: Thank you for the advice, we have added these references in our manuscript
While the primary motor cortex shows statistically significant differences between ALS patients and healthy controls, such differences do not reach the threshold of statistical significance when FDR correction is applied (Table 4). This observation probably deserves a bit of further discussion.
Answer: Thank you for the advice; we add the discussion in line 310-314.
In the comparison between ALS patients and healthy controls, I find that the choice of discarding 109 healthy controls and use only the 11 subjects whose age best matches that of patients is not very wise. To enable paired, age-matched comparison between patients and controls, the authors could devise a way to use synthetic normative data extracted from the fit. For example, for comparison with patient number 1, aged 58, authors could use “synthetic data” generated from their fit, taken at age = 58. This would enable to: (1) prevent the risk that the matched control subject is actually an outlier; (2) avoid to thrash the wealth of the information available from the 120 controls.
Answer: Thanks for the suggestion. Indeed, in this way one would have an ideal paired match between patients and controls. Unfortunately, not all ROIs reviled a significant age-correlation for measured values as shown in Fig.1B, Fig.2 and Table 1/2 so that we could/did not do it for all ROIs.
Finally, I have also the following relative minor comments:
The title is very non-informative. It doesn’t mention ALS at all. Further, the “purpose” section of the abstract does not make any mention of ALS, either. This is inconsistent with the current design of the paper, whose introduction (in the main text) begins as if the paper is principally about ALS.
Answer: We have now modified the abstract corresponding, to make it clear that the ALS patients were considered as an example of patients with neurodegenerative diseases (line 17).
In the methods, it is fine to motivate the choice for CAT12, but the details about the unused tools perhaps could be shortened.
Answer: We have now shortened the text about the unused tools accordingly.
In figures 1 A&B: I suggest to re-draw the plots of cortical thickness and keep the y-axis fixed (for example between 1.5 and 4) to visually depict the relative difference between the slopes.
Answer: Thanks for this advice. We re-draw Figures 1A&B and keep the y-axis fixed (between 1.65-3.9) as illustrated in the update manuscript.
A few typos are present throughout the paper, but these minor problems could be addressed first by the authors themselves during revision, and/or at a later stage of the review process.
Answer: Thank you for the comments; we have checked the typos again and tried to avoid any typo.
Reviewer 4 Report
This paper demonstrates a quantitative measurement of grey matter in the human brain. The authors investigated 120 healthy subjects and 11 patients with amyotrophic lateral sclerosis (ALS) by analyzing their T1-MPRAGE MR images with SPM12 software. The results showed the age-related reduction of cortical thickness/volume. ALS patients also showed a significant decrease in cortical thickness compared with healthy subjects.
The manuscript is well-written but such quantitative measurements of the brain cortex has already been used exclusively. These analyses are vigorously supported in SPM, FSL, Freesurfer, etc. The authors repeatedly mentioned that their results are consistent with the previous studies, suggesting that they had only reproduced these findings.
Otherwise I have no concerns on this manuscript.
Author Response
Comments and Suggestions for Authors
This paper demonstrates a quantitative measurement of grey matter in the human brain. The authors investigated 120 healthy subjects and 11 patients with amyotrophic lateral sclerosis (ALS) by analyzing their T1-MPRAGE MR images with SPM12 software. The results showed the age-related reduction of cortical thickness/volume. ALS patients also showed a significant decrease in cortical thickness compared with healthy subjects.
The manuscript is well-written but such quantitative measurements of the brain cortex has already been used exclusively. These analyses are vigorously supported in SPM, FSL, Freesurfer, etc. The authors repeatedly mentioned that their results are consistent with the previous studies, suggesting that they had only reproduced these findings.
Answer: Thank you for the comment. We agree that the quantitative measurements of the brain cortex have been used exclusively, but it is still changing to integrate the quantitative measurement in routine clinical diagnostic use. We just wanted to show that our results derived under clinical routine conditions are consistent with the previous studies and the qMR measurement may be applied under clinical routine condition as an add-on tool for diagnostics.
Otherwise I have no concerns on this manuscript.
Round 2
Reviewer 3 Report
In their Responses to Reviewers, the Authors have kindly replied to all my comments. However, it should be pointed out that two of my concerns (which I still consider “major” ones) were not addressed in the revised manuscript. The two issues are:
- The simple visual inspection of specific regional segmentation was not performed. Authors say that this will be done in a future paper.
- In the comparison between ALS patients and healthy controls, the Authors still discard 109 healthy controls and use only the 11 subjects whose age matched that of patients (+/- 2 years). I still believe that this approach is dangerous, because each chosen Control could be an outlier with respect to the fitted data. Authors replied that “Unfortunately, not all ROIs reviled a significant age-correlation for measured values shown in Fig.1B, Fig.2 and Table 1/2 so that we could/did not do it for all ROIs.” I would like to counter-reply that age dependency is not the major problem here: even if there is little or no age dependency in some ROIs, the individual patients should be compared to fitted data and not to one individual control subject.
In conclusion, my personal opinion is that Authors could have put a little bit more efforts in their revision. The significance and novelty of the manuscript is still relatively low, at least in its present form. The quality of presentation is acceptable.
Author Response
Comments and Suggestions for Authors
In their Responses to Reviewers, the Authors have kindly replied to all my comments. However, it should be pointed out that two of my concerns (which I still consider “major” ones) were not addressed in the revised manuscript. The two issues are:
1. The simple visual inspection of specific regional segmentation was not performed. Authors say that this will be done in a future paper.
Answer: We have done a very simple visual inspection of specific regional segmentation but not a detailed visual control for each ROI. That is the reason why we felt this is a weak point and described it as a limitation of the study. We have now inserted the word “detailed” to make this clear (line 328, in yellow shadow).
2. In the comparison between ALS patients and healthy controls, the Authors still discard 109 healthy controls and use only the 11 subjects whose age matched that of patients (+/- 2 years). I still believe that this approach is dangerous, because each chosen Control could be an outlier with respect to the fitted data. Authors replied that “Unfortunately, not all ROIs reviled a significant age-correlation for measured values shown in Fig.1B, Fig.2 and Table 1/2 so that we could/did not do it for all ROIs.” I would like to counter-reply that age dependency is not the major problem here: even if there is little or no age dependency in some ROIs, the individual patients should be compared to fitted data and not to one individual control subject.
Answer: As we mentioned last time that we would like to do the comparison as suggested. However, we could not derive fitted data for all ROIs. The reason is that we could not get a reasonable mathematic form for all selected ROIs. For the chosen values an outlier could be excluded based on our qualitative comparison (Figs. 3-4).
In conclusion, my personal opinion is that Authors could have put a little bit more efforts in their revision. The significance and novelty of the manuscript is still relatively low, at least in its present form. The quality of presentation is acceptable.
Answer: We have now modified the conclusion part as suggested with improved language style (line 339-340, in yellow shadow).